# Structural mechanisms of PIP$_2$ activation and SEA0400 inhibition in human cardiac sodium-calcium exchanger NCX1

Jing Xue[1,2,3†], Weizhong Zeng[1,2,3], Scott John[4], Nicole Attiq[5], Michela Ottolia[5]*, Youxing Jiang[1,2,3]*

[1]Department of Physiology, The University of Texas Southwestern Medical Center, Dallas, United States; [2]Department of Biophysics, The University of Texas Southwestern Medical Center, Dallas, United States; [3]Howard Hughes Medical Institute, Chevy Chase, United States; [4]Department of Medicine (Cardiology), UCLA, Los Angeles, United States; [5]Department of Anesthesiology and Perioperative Medicine, Division of Molecular Medicine, David Geffen School of Medicine, University of California Los Angeles, Los Angeles, United States

**\*For correspondence:** mottolia@g.ucla.edu (MO); youxing.jiang@utsouthwestern.edu (YJ)

**Present address:** †Renji Hospital, Shanghai Jiao Tong University School of Medicine, Shanghai, China

**Competing interest:** The authors declare that no competing interests exist.

**Sent for Review** 05 December 2024
**Preprint posted** 06 December 2024
**Reviewed preprint posted** 26 February 2025
**Reviewed preprint revised** 14 May 2025
**Version of Record published** 28 May 2025

## eLife Assessment

Cardiac Ca2+/Na+ exchange is mediated by the NCX1 antiporter, whose activity is tightly regulated. This **important** manuscript describes the structural basis of activation by the lipid DiC8-PIP2 and inhibition by binding of a small molecule to NCX1. These results provide **convincing** insights into NCX1 regulation and the structural basis of cellular Ca2+ signaling.

**Abstract** Na$^+$/Ca$^{2+}$ exchangers (NCXs) transport Ca$^{2+}$ across the plasma membrane in exchange for Na$^+$ and play a vital role in maintaining cellular Ca$^{2+}$ homeostasis. Our previous structural study of human cardiac NCX1 (HsNCX1) reveals the overall architecture of the eukaryotic exchanger and the formation of the inactivation assembly by the intracellular regulatory domain that underlies the cytosolic Na$^+$-dependent inactivation and Ca$^{2+}$ activation of NCX1. Here, we present the cryo-EM structures of HsNCX1 in complex with a physiological activator phosphatidylinositol 4,5-bisphosphate (PIP$_2$), or pharmacological inhibitor SEA0400, that enhances the inactivation of the exchanger. We demonstrate that PIP$_2$ binding stimulates NCX1 activity by inducing a conformational change at the interface between the transmembrane (TM) and cytosolic domains that destabilizes the inactivation assembly. In contrast, SEA0400 binding in the TM domain of NCX1 stabilizes the exchanger in an inward-facing conformation that facilitates the formation of the inactivation assembly, thereby promoting the Na$^+$-dependent inactivation of NCX1. Thus, this study reveals the structural basis of PIP$_2$ activation and SEA0400 inhibition of NCX1 and provides some mechanistic understandings of cellular regulation and pharmacology of NCX family proteins.

## Introduction

Sodium-calcium exchangers (NCXs) are transporters that control the flux of Ca$^{2+}$ across the plasma membrane and play a vital role in maintaining cellular calcium homeostasis for cell signaling (*Ottolia et al., 2021*; *Clapham, 2007*; *Blaustein and Lederer, 1999*; *Philipson and Nicoll, 2000*; *Berridge et al., 2003*; *DiPolo and Beaugé, 2006*). NCXs facilitate the exchange of three Na$^+$ for one Ca$^{2+}$ in an electrogenic manner, primarily responsible for extruding Ca$^{2+}$ from the cytoplasm. However, this

process can be reversed to permit $Ca^{2+}$ entry, depending on the chemical gradients of $Na^+$ and $Ca^{2+}$, as well as the membrane potential (*Blaustein and Lederer, 1999*; *Hilgemann et al., 1991*; *Reeves and Hale, 1984*; *Blaustein and Russell, 1975*; *Rasgado-Flores and Blaustein, 1987*; *Kimura et al., 1986*; *Matsuoka and Hilgemann, 1992*; *Kang and Hilgemann, 2004*). Three NCX isoforms (NCX1–3) are present in mammals, with each isoform bearing multiple splice variants expressed in distinct tissues, thereby modulating numerous fundamental physiological events (*Philipson and Nicoll, 2000*; *Philipson et al., 2004*; *Lee et al., 1994*; *Kofuji et al., 1994*; *Linck et al., 1998*; *Lytton, 2007*; *Dyck et al., 1999*). Dysfunctions of NCXs are associated with a plethora of human pathologies, including cardiac hypertrophy, arrhythmia, and postischemic brain damage (*Blaustein and Lederer, 1999*; *Watanabe et al., 2006*; *Pott et al., 2011*; *Matsuda et al., 1997*). The cardiac variant NCX1.1 has been extensively studied, with its function playing a central role in cardiac excitation and contractile activity (*Shigekawa and Iwamoto, 2001*; *Kimura et al., 1987*; *Bridge et al., 1990*; *Ottolia et al., 2013*; *Scranton et al., 2024*).

The eukaryotic NCX consists of a transmembrane (TM) domain with 10 TM helices and a large intracellular regulatory domain between TMs 5 and 6 (*Philipson and Nicoll, 2000*; *Ren and Philipson, 2013*; *Sharma and O'Halloran, 2014*; *Matsuoka et al., 1993*; *Hilgemann, 1990*; *Xue et al., 2023*; *Dong et al., 2024*). The TM domain is responsible for the ion exchange function in NCX. It consists of two homologous halves (TMs 1–5 and TMs 6–10) with TMs 2–3 and TMs 7–8 forming the core of the TM domain and hosting the residues that coordinate the transported $Na^+$ and $Ca^{2+}$ (*Liao et al., 2012*; *Liao et al., 2016*). The large regulatory domain contains two calcium-binding domains (CBD1 and CBD2). $Ca^{2+}$ binding at CBDs enhances NCX activity and rescues it from the $Na^+$-dependent inactivation, a process that manifests as slow decay of the exchange current due to elevated levels of cytosolic $Na^+$ ions (*Matsuoka and Hilgemann, 1992*; *Matsuoka et al., 1993*; *Hilgemann, 1990*; *Matsuoka et al., 1995*; *Hilge et al., 2006*; *Ottolia et al., 2009*; *Hilgemann et al., 1992a*; *Hilgemann et al., 1992c*; *Matsuoka and Hilgemann, 1994*). A stretch of residues known as the XIP region (eXchanger Inhibitory Peptide) at the N-terminus of the regulatory domain plays a pivotal role in the $Na^+$ inactivation process (*Matsuoka et al., 1997*; *Li et al., 1991*).

Several other cellular cues can also modulate NCX1 activity, including phosphatidylinositol 4,5-bisphosphate ($PIP_2$) in the membrane. $PIP_2$ has been shown to stimulate the NCX1 activity by reducing the $Na^+$-dependent inactivation, and the XIP region was suggested to participate in the $PIP_2$ activation (*Hilgemann et al., 1992a*; *He et al., 2000*; *Hilgemann and Ball, 1996*; *Yaradanakul et al., 2007*). In addition, several small molecule NCX inhibitors have been developed to provide valuable tools for studying the physiological and pharmacological properties of NCXs, among which compound SEA0400 is a highly potent and selective NCX1 inhibitor that promotes the $Na^+$-dependent inactivation of the exchanger (*Watanabe et al., 2006*; *Tanaka et al., 2002*; *Matsuda et al., 2001*; *Watano et al., 1996*; *Lee et al., 2004*; *Bouchard et al., 2004*; *Iwamoto et al., 2004*).

We previously determined the human cardiac NCX1 structure in an inward-facing inactivated state in which XIP and the β-hairpin between TMs 1 and 2 form a TM-associated four-stranded β-hub and mediate a tight packing between the TM and cytosolic domains, resulting in the formation of a stable inactivation assembly that blocks the TM movement required for ion exchange function (*Xue et al., 2023*). The study also provides mechanistic insight into how cytosolic $Ca^{2+}$ binding at CBD2 destabilizes the inactivation assembly and activates the exchanger (*Xue et al., 2023*). To expand our understanding of NCX modulation by $PIP_2$ lipid and small molecule inhibitors, we present the cryo-EM structures of human cardiac NCX1 in complex with $PIP_2$ or SEA0400, revealing the structural basis underlying $PIP_2$ activation and SEA0400 inhibition of NCX1.

## Results

### $PIP_2$ activation of NCX1

$PIP_2$ has been shown to modulate NCX1 activity by reducing the $Na^+$-dependent inactivation of the exchanger (*Hilgemann et al., 1992a*; *He et al., 2000*; *Hilgemann and Ball, 1996*; *Yaradanakul et al., 2007*). To characterize the effect of $PIP_2$ on HsNCX1, we expressed the exchanger in *Xenopus laevis* oocytes and recorded the outward exchanger currents using the giant patch technique in the inside-out configuration with or without applying porcine brain $PIP_2$ (*Figure 1* and Methods). The recording was performed with 12 μM free cytosolic $[Ca^{2+}]_i$ (bath), and the outward NCX1 current was

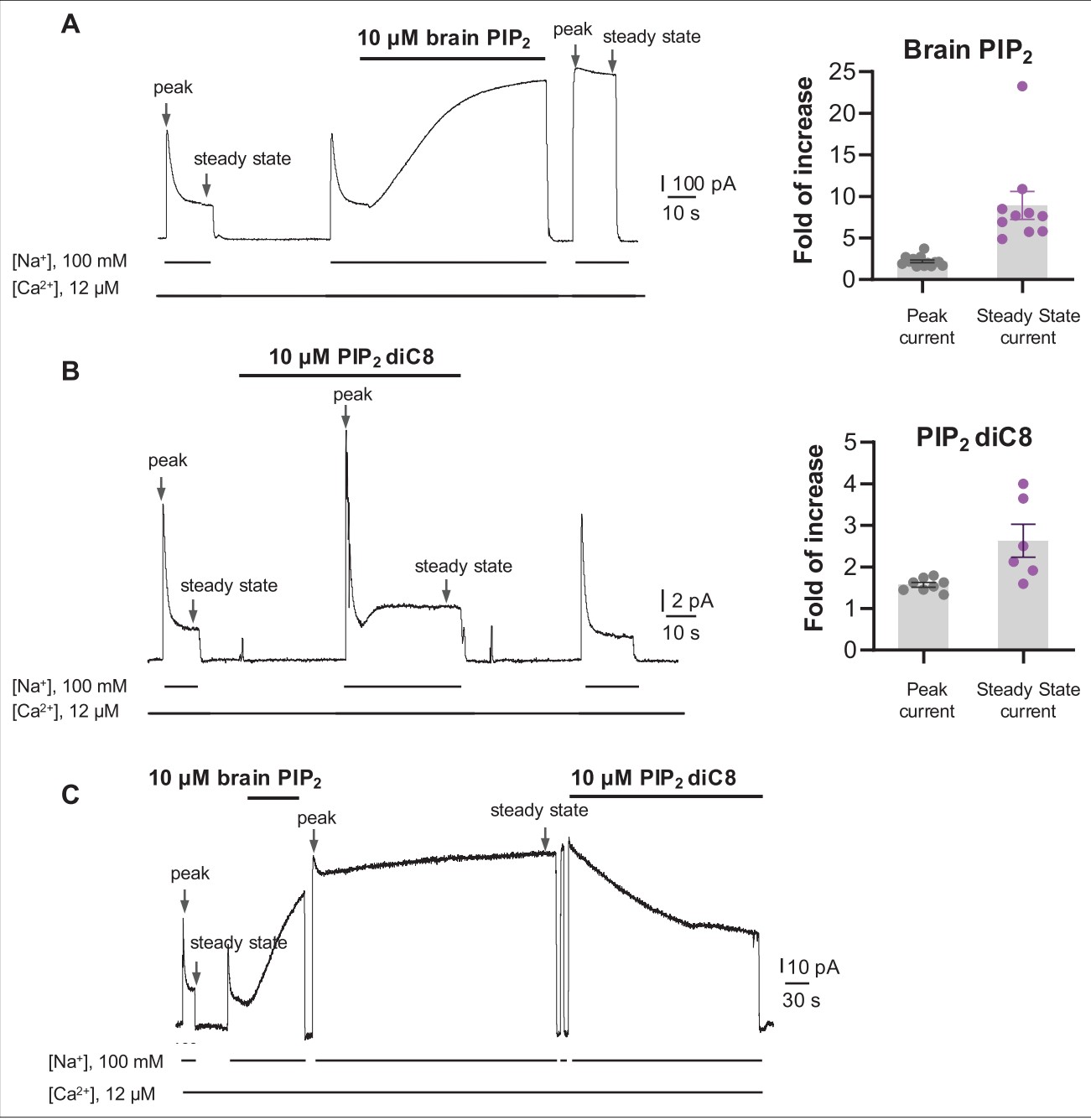

**Figure 1.** Phosphatidylinositol 4,5-bisphosphate (PIP₂) enhances HsNCX1 activity. (**A**) Representative outward currents recorded from oocytes expressing the human NCX1 before and after application of long-chain brain PIP₂. Currents were activated by replacing cytosolic Cs⁺ with Na⁺. Application of 10 µM brain PIP₂ enhanced HsNCX1 current and abolished the Na⁺-dependent inactivation irreversibly. Perfusion time of PIP₂ is indicated above traces, while lines below traces indicate solution exchange. Arrows mark the peak and steady currents used to measure the fold of increase upon PIP₂ application. The fold of current increase was calculated by comparing the peak or steady-state current before and after PIP₂ application (fold of increase in peak current = 2.2 ± 0.2, n=14; steady-state current = 8.9 ± 1.7, n=10; data points are mean ± s.e.m.). (**B**) Representative outward currents recorded before and after application of short-chain PIP₂ diC8 (10 µM). PIP₂ diC8 was perfused from the cytosolic side before HsNCX1 activation (in the presence of Cs⁺ for 30 s) and during transport (in the presence of Na⁺). Both peak and steady-state currents of HsNCX1 are enhanced by PIP₂ diC8, and the effect is reversible (fold of current increase in peak current = 1.6 ± 0.1, n=8; steady-state current = 2.6 ± 0.4, n=6; data points are mean ± s.e.m.). The Na⁺-dependent inactivation remains in the presence of PIP₂ diC8. (**C**) Representative outward currents recorded with the application of brain PIP₂ and PIP₂ diC8. The NCX1 current was first potentiated by applying 10 µM brain PIP₂ at the steady state. The PIP₂ effect was not reversible over the 5 min washout with a solution containing 100 mM Na⁺ and 12 µM Ca²⁺. The same patch was then perfused with the same solution in the presence of 10 µM PIP₂ diC8.

*Figure 1 continued on next page*

**Figure 1 continued**

Application of the short-chain $PIP_2$ diC8 facilitates the decrease of brain $PIP_2$-potentiated current, suggesting that both lipids compete for the same binding site.

The online version of this article includes the following source data for figure 1:

**Source data 1.** The fold of current increase in the peak or steady-state current before and after $PIP_2$ application.

elicited by the rapid replacement of 100 mM $Cs^+$ with 100 mM $Na^+$ in the bath solution. In the patches without applying $PIP_2$, the outward exchanger currents quickly decay and reach a steady state due to $Na^+$-dependent inactivation (*Figure 1A*). Introducing 10 µM $PIP_2$ at the steady state progressively increases the current that plateaus at about twofold of the peak current measured before $PIP_2$ addition (*Figure 1A*). After $PIP_2$ washout, the outward current remains at the pre-removal level without obvious inactivation, indicating high-affinity $PIP_2$ binding and its positive modulation of NCX1 by both potentiating the peak current and reducing the $Na^+$-dependent inactivation. Intriguingly, the commonly used shorter chain $PIP_2$ substitute ($PIP_2$ diC8) does not have the equivalent activation effect on NCX1, and the exchanger remains susceptible to $Na^+$-dependent inactivation when recorded in the presence of 10 µM $PIP_2$ diC8 (*Figure 1B*). However, $PIP_2$ diC8 still binds and stimulates both the peak and steady currents of the exchanger. This stimulation effect is abolished after $PIP_2$ diC8 washout, indicating a lower affinity of short-chain $PIP_2$ than that of the long-chain native $PIP_2$.

To verify that the short-chain $PIP_2$ diC8 and the long-chain brain $PIP_2$ share the same binding site, we performed a competition assay. As shown in *Figure 1C*, introducing high-affinity brain $PIP_2$ at the steady state yields a long-lasting potentiation of NCX1 current that is irreversible even after a 5 min washout. Applying $PIP_2$ diC8 can steadily decrease the brain $PIP_2$-potentiated NCX1 current, suggesting that both lipids compete for the same binding site.

## Structural insight into $PIP_2$ binding in NCX1

To reveal the structural mechanism of $PIP_2$ activation, we tried to obtain the EM structure of HsNCX1 in the presence of the long-chain porcine brain $PIP_2$. However, the exchanger becomes highly dynamic, yielding a low-resolution EM map with an overall shape similar to a cytosolic $Ca^{2+}$-activated NCX1 whose β-hub-mediated inactivation assembly is destabilized and cytosolic domain (CBD1 and CBD2) is detached from the TM domain (*Xue et al., 2023*: *Figure 2—figure supplement 1*). We suspect the long-chain $PIP_2$ exerts the same activation effect on NCX1 as high cytosolic $Ca^{2+}$ by destabilizing the inactivation assembly. As NCX1 retains its $Na^+$-dependent inactivation property in the presence of the short-chain $PIP_2$, we reasoned that the $PIP_2$ diC8-bound NCX1 likely remains in an inward-facing inactivated state in high $Na^+$ low $Ca^{2+}$ condition, and its structure would still reveal how $PIP_2$ binds in NCX1. We therefore determined the EM structure of NCX1 in complex with $PIP_2$ diC8 at 3.5 Å (*Figure 2*, *Figure 2—figure supplement 2*, *Table 1*, and Methods), which indeed adopts an inward-facing conformation with intact inactivation assembly. Due to the relative dynamic movement between the TM and cytosolic domains, we also performed local refinement to improve the map quality for each domain (*Figure 2—figure supplement 2*). The density of the $IP_3$ head group from the bound $PIP_2$ diC8 is well defined in the local-refined EM map focused on the TM domain (*Figure 2A*, *Figure 2—figure supplement 2B*). This density is not present in the apo NCX1 structure (*Figure 2—figure supplement 3*). The acyl chains, however, are flexible and could not be resolved in the structure (*Figure 2—figure supplement 2*). The lipid is attached to the cytosolic sides of TMs 4 and 5 with its head group positioned at the C-terminal end of TM5 (*Figure 2A*). Four positively charged residues, including K164 and R167 from the N-terminus of TM4 and R220 and K225 from the C-terminus of TM5, are positioned in the vicinity of the $PIP_2$ head group and likely participate in the electrostatic interactions with the head group (*Figure 2A*).

## $PIP_2$ diC8-induced conformational changes in NCX1

Two major conformational changes occur in NCX1 upon $PIP_2$ diC8 binding (*Figure 2B–D*). The first change occurs at the C-terminus of TM5, which ends at R220 and is connected to the two-stranded XIP β-sheet (β3 and β4) via a six-residue loop in the apo structure. When $PIP_2$ binds, the TM5 helix extends by one helical turn. This loop-to-helix transition significantly changes the locations of these connecting loop residues and their side-chain orientations to accommodate $PIP_2$, enabling K225 to

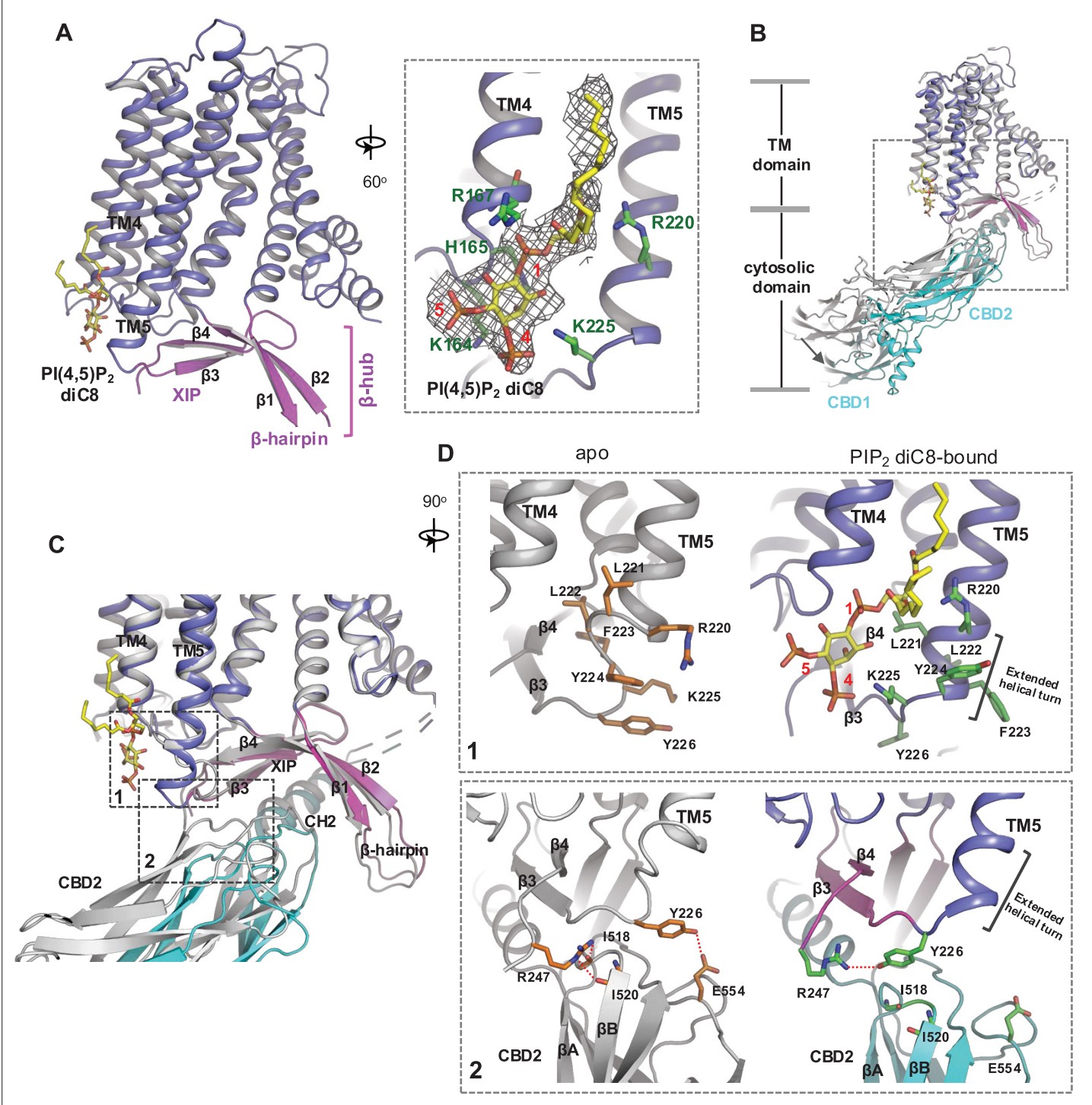

**Figure 2.** Phosphatidylinositol 4,5-bisphosphate (PIP$_2$) binding in NCX1. (**A**) Structure of the transmembrane (TM) and β-hub regions of PIP$_2$ diC8-bound NCX1 with a zoomed-in view of the lipid-binding site. The density of PIP$_2$ diC8 is shown as a gray mesh contoured at 5.5σ. (**B**) Structural comparison between apo (gray) and PIP$_2$ diC8-bound (color) NCX1. Upon PIP2 diC8 binding, there is a rigid-body downward swing movement (marked by an arrow) at CBDs caused by the partial detachment of the CBD2 domain from XIP. The conformational change at the TM domain is subtle and mainly occurs at the C-terminus of TM5 as illustrated in (**C**) and (**D**). (**C**) Zoomed-in view of the structural comparison (boxed area in (**B**)). The two major conformational changes occur in the boxed regions. (**D**) Zoomed-in views of the two conformational changes between apo (left in gray) and PIP$_2$ diC8-bound (right in color) state. Top: conformational change 1 at the C-terminus of TM5. Bottom: conformational change 2 at the interface between XIP and CBD2.

The online version of this article includes the following figure supplement(s) for figure 2:

*Figure 2 continued on next page*

*Figure 2 continued*

**Figure supplement 1.** Cryo-EM data processing of HsNCX1 in the presence of long-chain porcine brain phosphatidylinositol 4,5-bisphosphate (PIP$_2$).

**Figure supplement 2.** Structure determination of HsNCX1-PIP$_2$ diC8 complex.

**Figure supplement 3.** Side-by-side comparison of the densities at the phosphatidylinositol 4,5-bisphosphate (PIP$_2$)-binding site between the PIP$_2$-bound structure (EMD-60921) and the apo structure (EMD-40457).

**Figure supplement 4.** Proposed structural basis underlying the different binding affinity between long- and short-chain phosphatidylinositol 4,5-bisphosphate (PIP$_2$).

**Table 1.** Cryo-EM data collection and model statistics.

| Sample preparation conditions | 25 mM HEPES pH 7.4, 200 mM NaCl 0.9 mM SEA0400 | 25 mM HEPES pH 7.4, 200 mM NaCl, 0.47 mM PI(4,5)P$_2$diC8 |
|---|---|---|
| | SEA0400-bound state (EMD-40456, PDB 8SGI) | PI(4,5)P$_2$diC8-bound state (EMD-60921, PDB 9IV8) |
| **Data collection and processing** | | |
| Magnification | 105k | 105k |
| Voltage (kV) | 300 | 300 |
| Electron exposure (e$^-$/Å$^2$) | 60 | 60 |
| Defocus range (μm) | –0.9 to –2.2 | –0.9 to –2.2 |
| Pixel size (Å) | 0.83 | 0.84 |
| Symmetry imposed | C1 | C1 |
| Initial particle images (no.) | 1,249,151 | 2,233,044 |
| Final particle images (no.) | 368,227 | 117,748 |
| Map resolution (Å) FSC threshold | 2.93 0.143 | 3.47 0.143 |
| **Refinement** | | |
| Initial model used (PDB code) | 8SGJ | 8SGJ |
| Model resolution (Å) FSC threshold | 3.36 0.5 | 3.81 0.5 |
| Model composition Non-hydrogen atoms Protein residues Ligands | 7667 982 3: Na 6: Ca 1: H$_2$O 1: SEA0400 | 5947 750 5: Ca 1: PI(4,5)P$_2$diC8 |
| B factors (Å$^2$) Protein Ligands | 66.21 58.55 | 50.47 106.20 |
| R.m.s. deviations Bond lengths (Å) Bond angles (°) | 0.005 0.696 | 0.004 0.679 |
| Validation MolProbity score Clashscore Poor rotamers (%) | 1.36 5.24 0 | 1.28 5.22 0 |
| Ramachandran plot Favored (%) Allowed (%) Disallowed (%) | 97.62 2.38 0 | 98.24 1.76 0 |

reorient and interact with the IP$_3$ head group (*Figure 2D*, top panel). The second conformational change is the partial detachment of the CBD2 domain from XIP upon PIP$_2$ binding, resulting in a downward swing of cytosolic CBD domains (CBD1 and CBD2) as a rigid body (*Figure 2B and C*). This CBD2 detachment is a result of the first PIP$_2$-induced conformational change at TM5 that disrupts part of the interactions between CBD2 and XIP as follows: In the apo-inactivated state, Y226 and R247, the two termini residues of the two-stranded XIP β-sheet, form H-bonds with several CBD2 residues, including E554 side chain and backbone carbonyls of I518 and I520 (*Figure 2D*, bottom panel). The loop-to-helix transition of TM5 upon PIP$_2$ binding leads to a dramatic rotation of Y226 that allows it to move closer to and directly interact with R247, resulting in the loss of H-bonding interactions between XIP and CBD2 (*Figure 2D*, bottom panel). In addition, the rotation of Y226 also causes a direct collision with CBD2 if it remains closely attached to XIP. Thus, the PIP$_2$-induced TM5 movement, particularly the reorientation of Y226, abolishes some local interactions between CBD2 and XIP and also pushes CBD2 away from XIP, causing the partial detachment of CBD2. However, the short-chain PIP$_2$ only partially destabilizes rather than completely disassembles the inactivation assembly, as the CH2 helix of CBD2 still engages in extensive interactions with the C-shaped β-hub.

To test if the PIP$_2$-interacting residues play a critical role in the native long-chain PIP$_2$ activation, we performed mutagenesis to those positively charged residues, including K164A, R167A, R220A, and K225A single mutants. We also generated an R220A/K225A double mutant as these two residues undergo PIP$_2$-induced conformational change at TM5. While all mutants remain susceptible to current potentiation upon brain PIP$_2$ application, as seen in the wild-type NCX1, the PIP$_2$ potentiation effects on peak and steady-state currents are weakened in some mutants, most notably in R220A and R220A/K225A (*Figure 3A–C*). Interestingly, these two mutants also have reduced Na$^+$-dependent inactivation as demonstrated by their higher fractional activity (ratio between steady-state and peak currents) before applying PIP$_2$ (*Figure 3D*). The retained PIP$_2$ activation on these mutants implies that the longer acyl chain from native PIP$_2$ plays an important role in its interaction and activation of NCX1. Furthermore, as the interactions between PIP$_2$ and NCX1 are both electrostatic involving multiple charged residues and hydrophobic involving the long lipid acyl chain, those single or double amino acid substitutions may only decrease the affinity of PIP$_2$ rather than abolish its binding. To test that, we also mutated all four positively charged residues to alanine. The K164A/R167A/R220A/K225A mutant is no longer sensitive to PIP$_2$ activation. The currents from this quadruple mutant are small in most recordings and show no Na$^+$-dependent inactivation. The unresponsiveness to PIP$_2$ and lack of Na$^+$-dependent inactivation in this mutant is consistent with previous studies, demonstrating that PIP$_2$ activates NCX by tuning the amount of Na$^+$-dependent inactivation, and any mutation that decreases NCX sensitivity to PIP$_2$ will also affect the extent of Na$^+$-dependent inactivation (*He et al., 2000*). This quadruple NCX1 mutant likely abolishes the potentiation effect from both endogenous and externally applied PIP$_2$ and thereby functions in a Na$^+$-inactivated steady state.

## Structure of NCX1 in complex with SEA0400 inhibitor

SEA0400 is known to potently inhibit cardiac NCX1 by facilitating the inactivation of the exchanger (*Lee et al., 2004*; *Bouchard et al., 2004*; *Iwamoto et al., 2004*). To reveal the structural mechanism of its inhibition, we determined the structure of HsNCX1 in complex with SEA0400 at 2.9 Å (*Figure 4*, *Figure 4—figure supplement 1*, *Table 1*, and Methods). The SEA0400-bound NCX1 structure adopts an inward-facing, inactivated state identical to the apo NCX1 structure obtained at high Na$^+$, nominal Ca$^{2+}$-free condition (*Xue et al., 2023*: *Figure 4A*, *Figure 4—figure supplement 2*). SEA0400 binds at the TM domain in a pocket enclosed by the internal halves of TMs 8 (8a segment), 2 (2ab segments), 4, and 5 (*Figure 4B*). The pocket has a lateral fenestration in the middle of the membrane (*Figure 4B*), which provides a portal for the SEA0400 entrance. The pocket is sealed off at the cytosolic side by E244 from the XIP β-sheet. As demonstrated in our previous study (*Xue et al., 2023*), XIP and the linker β-hairpin (β1 and β2) between TMs 1 and 2ab form a β-hub that stabilizes the exchanger in an inactivated state. This β-hub has to be disassembled in an activated NCX1, and XIP is expected to be detached from the TM domain, which would lead to the opening of the SEA400-binding pocket to the cytosol and provide a cytosolic portal for the release of the inhibitor, as further discussed below. *Figure 4C* summarizes the interactions between SEA0400 and NCX1 and demonstrates that SEA0400 fits perfectly in the pocket, making extensive contact with surrounding residues. Mutations of some key interacting residues, such as F213 and G833, have been shown to compromise the inhibitor

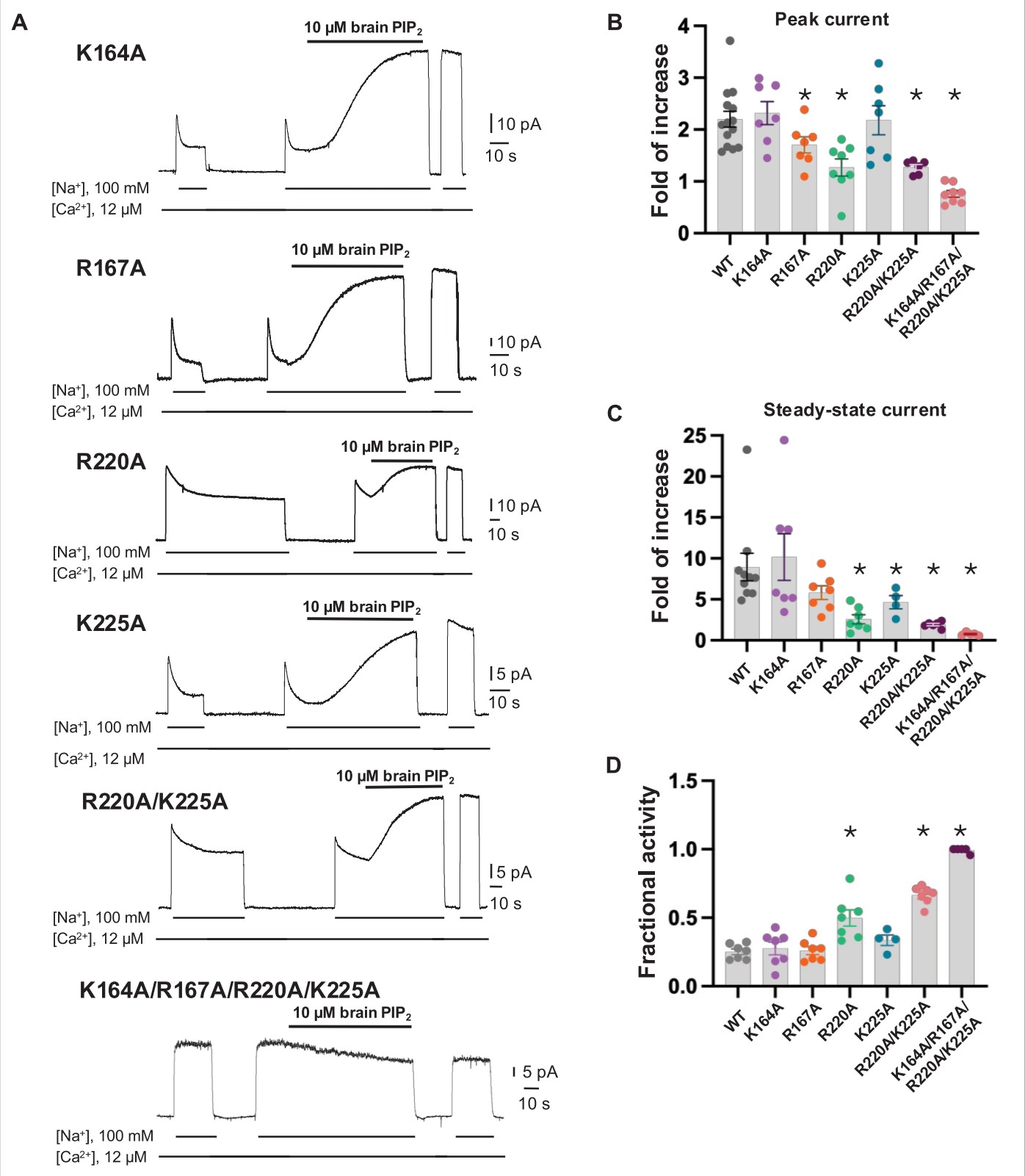

**Figure 3.** Mutagenesis at the phosphatidylinositol 4,5-bisphosphate (PIP$_2$) binding. (**A**) Representative NCX1 currents of PIP$_2$ site mutants before and after perfusion of 10 µM brain PIP$_2$ to the cytosolic side of the patch. (**B, C**) Summary graphs demonstrating the effects of PIP$_2$ on the enhancement of peak (**B**) and steady-state currents (**C**). Potentiation (fold of increase) was measured by comparing the current magnitude before and after PIP$_2$ application. Mutants R167A, R220A, and K225A showed some decreased response to PIP$_2$, whereas the R220A/K225A mutant shows a more profound

*Figure 3 continued on next page*

*Figure 3 continued*

decrease in PIP$_2$ response. Compared to WT, the PIP$_2$ potentiation of R220A/K225A at the steady state is decreased by ~70–90% (fold of increase WT = 8.9 ± 1.6, n=10 vs R220A/K225A=1.9 ± 0.1, n=6). PIP2 has no potentiation effect on the quadruple K164A/R167A/R220A/K225A mutant. Data points are mean ± s.e.m. (*p<0.1). (**D**) The extent of Na$^+$-dependent inactivation was measured as the ratio between steady-state and peak currents (fractional activity), and values for WT and the indicated mutants are shown. Mutants R220A and R220A/K225A displayed significantly higher fractional activity values when compared to WT, indicating that the Na$^+$-dependent inactivation was less pronounced in these mutant exchangers. K164A/R167A/R220A/K225A shows no Na$^+$-dependent inactivation.

The online version of this article includes the following source data for figure 3:

**Source data 1.** Fold of increase in peak and steady-state current, and fractional activity values for WT and indicated mutants.

binding (*Iwamoto et al., 2004*). The same SEA0400 binding was also demonstrated in the recent study of human NCX1.3, and mutations at some pocket-forming residues mitigate the SEA0400 inhibition of the exchanger (*Dong et al., 2024*).

## Inhibition mechanism of SEA0400

The TM domain of NCX1 shares a similar overall architecture to the archaea exchanger NCX_Mj (*Liao et al., 2012*). The structural comparison between the TM domains of the inward-facing NCX1 and the outward-facing NCX_Mj reveals the conformational changes that occur during ion exchange, providing structural insight into the SEA0400 inhibition mechanism (*Figure 5A*). The inward-outward transition mainly involves the sliding motion of TMs 1 and 6 and the bending movement of TMs 2ab and 7ab (*Xue et al., 2023*; *Liao et al., 2012*; *Liao et al., 2016*; *Marinelli and Faraldo-Gomez, 2023*). As TM2ab directly interacts with SEA0400 in the inward-facing state, its bending movement toward the outward conformation would cause a direct collision with the inhibitor (*Figure 5A*). Thus, the binding of SEA0400 stabilizes the exchanger in the inward-facing state and blocks the conformational change from the inward to the outward state. The Na$^+$-dependent NCX1 inactivation occurs when the exchanger is in a Na$^+$-loaded, inward-facing state with low cytosolic [Ca$^{2+}$] (*Hilgemann et al., 1992c*; *Matsuoka and Hilgemann, 1994*; *Hilgemann, 2020*). As demonstrated in our previous study, only in this state can the β-hub form and readily interact with the cytosolic CBD domains, generating the inactivation assembly that locks TMs 1 and 6 and prevents the TM module from transporting ions (*Xue et al., 2023*). Thus, SEA0400 promotes NCX1 inactivation by stabilizing NCX1 in the inward-facing conformation, which facilitates the formation of the inactivation assembly. The formation of the inactivation assembly also reciprocally stabilizes SEA0400 binding as the XIP of the assembly interacts with the TM domain and seals off the inhibitor binding pocket from the cytosolic side (*Figure 5B*). Under conditions in which the inactivation assembly cannot form in NCX1, such as chymotrypsin treatment or forward exchange mode (Na$^+$ influx/Ca$^{2+}$ efflux), the removal or detachment of XIP from the TM domain would generate a cytosolic open portal for SEA0400 release and effectively reduce its binding affinity (*Figure 5C*; *Lee et al., 2004*; *Bouchard et al., 2004*). Indeed, cysteine scanning mutagenesis studies have shown that the pocket-forming G833 residue is accessible to intracellular sulfhydryl reagents in the chymotrypsin-treated inward-facing NCX1, confirming the cytosolic exposure of the SEA400-binding pocket upon XIP removal (*John et al., 2013*). This cytosolic opening of the SEA0400 pocket explains the low efficacy of SEA0400 inhibition in NCX1 when the Na$^+$-dependent inactivation is absent.

## Discussion

In this study, we provide mechanistic underpinnings of cellular regulation and pharmacology of NCX family proteins by revealing the structural basis of PIP$_2$ activation and SEA0400 inhibition in HsNCX1. Both compounds modulate NCX1 activity by reducing or enhancing the physiologically relevant Na$^+$-dependent inactivation process that occurs when the exchanger is in an inward-facing, Na$^+$-loaded state with high [Na$^+$] and low [Ca$^{2+}$] on the cytosolic side (*Scranton et al., 2024*; *Hilgemann et al., 1992c*). In this inactivation state, XIP and β-hairpin can assemble into a TM-associated four-stranded β-hub that mediates a tight packing between the TM and cytosolic domains, resulting in the formation of an inactivation assembly that blocks the TM conformational changes required for ion exchange function (*Xue et al., 2023*; *Dong et al., 2024*). PIP$_2$ or SEA0400 binding changes the stability of the inactivation assembly in NCX1, resulting in a reduction or potentiation of inactivation.

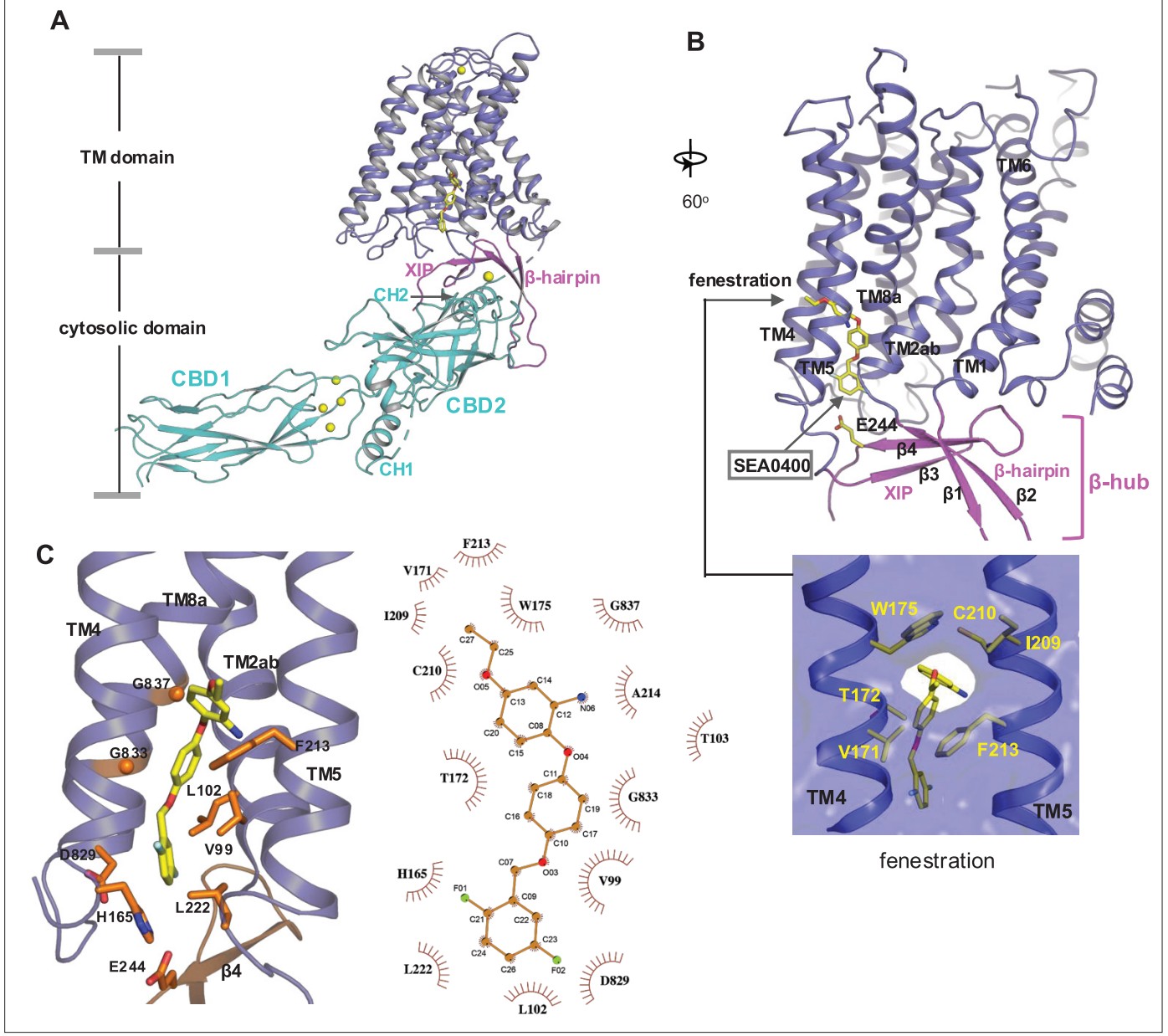

**Figure 4.** SEA0400 binding in NCX1. (**A**) Overall structure of human NCX1 in complex with SEA0400 obtained in high Na+ and low Ca2+ conditions. Yellow spheres represent the bound Ca2+ in CBD1 and XIP. (**B**) Cartoon representation of the transmembrane (TM) domain and β-hub of the complex with surface-rendered view of the fenestration in the middle of the membrane. The β-hub is assembled by β-hairpin (β1 and β2) and XIP (β3 and β4). (**C**) Zoomed-in view of the SEA0400-binding site, and the schematic diagram detailing the interactions between NCX1 residues and SEA0400.

The online version of this article includes the following figure supplement(s) for figure 4:

**Figure supplement 1.** Structure determination of HsNCX1-SEA0400 complex.

**Figure supplement 2.** Structural comparison between the apo (gray) and SEA0400-bound (color) HsNCX1.

Although short-chain PIP$_2$ cannot fully recapitulate the NCX1 activation effect from long-chain native PIP$_2$, the structure of PIP$_2$ diC8 NCX1 allows us to define the lipid-binding site and the lipid-induced conformational changes at the interface between CBD2 and XIP that destabilize the inactivation assembly. The SEA0400-bound NCX1 structure presented here, along with the recent study by *Dong et al., 2024*, suggests that the drug can directly inhibit NCX1 by blocking the inward-outward conformational change at TM2ab. Our structural analysis also explains the strong connection between SEA0400 binding and Na+-dependent inactivation - SEA0400 is ineffective in an exchanger lacking

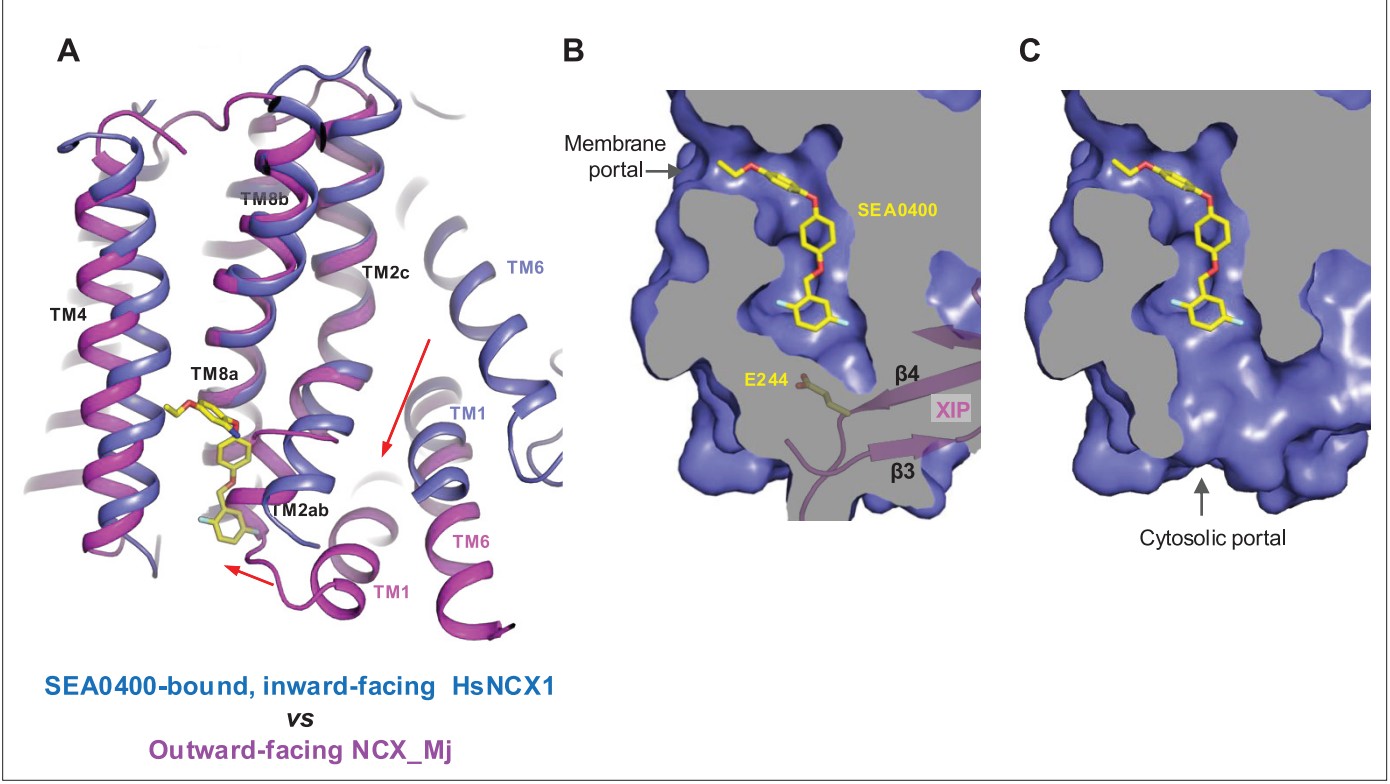

**Figure 5.** Structural mechanism of SEA0400 inhibition. (**A**) Structural comparison at the core part of the transmembrane (TM) domain between the SEA0400-bound, inward-facing HsNCX1 and the outward-facing NCX_Mj (PDB 3V5U). Red arrows mark the sliding movement of TMs 1 and 6 and the bending of TM2ab from inward to outward conformation. (**B**) Surface-rendered views of the SEA0400-binding pocket sealed off from the cytosolic side by E244 from XIP in the inactivated state. (**C**) Removal of XIP would generate a cytosolic portal that facilitates the release of SEA0400.

Na$^+$-dependent inactivation, whereas enhancing the extent of Na$^+$-dependent inactivation increases the affinity for SEA0400. As SEA0400 binding traps NCX1 in an inward-facing state that facilitates the formation of the inactivation assembly, the interaction between XIP and the TM domain in the assembly can in turn stabilize SEA0400 binding by sealing the drug-binding pocket from the cytosol. When XIP is removed, as in the chymotrypsin-treated NCX1, or when the inactivation assembly cannot form, as in the NCX1 at forward exchange mode, the SEA0400-binding pocket becomes exposed to the cytosol, mitigating the inhibition efficacy of SEA0400 (*Lee et al., 2004*; *Bouchard et al., 2004*).

While we expect the long-chain PIP$_2$ to bind in the same location as PIP$_2$ diC8, it is unclear how it exacerbates the destabilization effect on the inactivation assembly. The long acyl chain of native PIP$_2$ may engage in some interactions with the TM module of NCX1 that are not present in the short-chain lipid, rendering a higher affinity binding and more profound destabilization effect on the inactivation assembly than PIP$_2$ diC8. The acyl-chain length-dependent PIP$_2$ activation is consistent with some previous studies. Before PIP$_2$ was demonstrated to regulate NCX, some earlier studies showed that negatively charged long-chain lipids such as phosphatidylserine or phosphatidic acid could have the same potentiation effects on NCX1 as PIP$_2$ (*Hilgemann and Collins, 1992b*; *Vemuri and Philipson, 1988*). Furthermore, long-chain acyl-CoAs could also have the same potentiation effects on NCX as PIP$_2$ (*Riedel et al., 2006*). All these studies demonstrated that activation of NCX by the anionic lipids depends on their chain length, with the short chain being ineffective or less effective. These findings have two implications. First, it is the negative surface charge rather than the specific IP$_3$ head group of the lipid that is important for stimulating NCX1 activity, suggesting nonspecific electrostatic interactions between the negatively charged lipids and those positively charged residues at the binding site. Second, a longer acyl chain is required for the high-affinity binding of PIP$_2$ or negatively charged lipids. In the PIP$_2$ diC8-bound structure, the tail of the acyl chain is positioned right at the fenestration of NCX1 that serves as the portal for the SEA0400 binding. Two pieces of evidence lead us to suggest that the tail of the long acyl chain from a native lipid can enter the same binding pocket for

SEA0400 and thereby render its higher affinity binding than a shorter chain lipid. First, a docking analysis (*Eberhardt et al., 2021*; *Bugnon et al., 2024*) of both long-chain and short-chain PIP$_2$ at the binding site showed that the tail portion of the acyl chain from the native brain PIP$_2$ inserts into the SEA0400-binding pocket through the fenestration of NCX1 in all docking models with the highest calculated affinity (*Figure 2—figure supplement 4A*). This pocket is not reachable for PIP$_2$ diC8 due to its shorter chain length (*Figure 2—figure supplement 4B*). Second, a stretch of density likely from a native lipid acyl chain is observed in the apo NCX1 structure inside the SEA0400-binding pocket (*Figure 2—figure supplement 4C*), indicating its accessibility to a long-chain lipid. Thus, the accommodation of the long acyl chain in the open pocket of NCX1 through the fenestration in the middle of the membrane likely contributes to the high-affinity binding of native lipids.

## Methods

**Key resources table**

| Reagent type (species) or resource | Designation | Source or reference | Identifiers | Additional information |
|---|---|---|---|---|
| Gene (*Homo sapiens*) | NCX1 (sodium/calcium exchanger 1) | Uniprot | P32418 | |
| Strain, strain background (*Escherichia coli*) | TOP10 | Thermo Fisher Scientific | Cat# 18258012 | |
| Strain, strain background (*E. coli*) | DH10Bac | Thermo Fisher Scientific | Cat# 10361012 | |
| Cell line (*Spodoptera frugiperda*) | Sf9 cells | Thermo Fisher Scientific | Cat# 11496015; RRID:CVCL_0549 | |
| Cell line (*H. sapiens*) | Expi293 GnTI- Cells | Thermo Fisher Scientific | Cat# A39240; RRID:CVCL_B0J7 | |
| Transfected construct (*H. sapiens*) | pEZT-BM-NCX1-C$_{strep}$ | This paper | N/A | |
| Recombinant DNA reagent | pEZT-BM | *Morales-Perez et al., 2016* | RRID:Addgene_74099 | |
| Sequence-based reagent | NCX1_F_primer | This paper | PCR primers | gtacttaatacgactcactataggctagcgccaccatgta caacatgcggcgattaagtc |
| Sequence-based reagent | NCX1_R_primer | This paper | PCR primers | gatggctccatgagccaccAgcggccgcgaagccttttat gtggcagtaggc |
| Chemical compound, drug | Sodium Butyrate | Sigma-Aldrich | Cat# 303410 | |
| Chemical compound, drug | Lauryl Maltose Neopentyl Glycol | Anatrace | Cat# NG310 | |
| Chemical compound, drug | Digitonin | Acros Organics | Cat# 11024-24-1 | |
| Chemical compound, drug | Biotin | Sigma-Aldrich | Cat# B4501 | |
| Software, algorithm | MotionCor2 | *Zheng et al., 2017* | RRID:SCR_016499 | |
| Software, algorithm | GCTF | *Zhang, 2016* | RRID:SCR_016500 | |
| Software, algorithm | RELION | *Scheres, 2012* | RRID:SCR_016274 | http://www2.mrc-lmb.cam.ac.uk/relion |

*Continued on next page*

*Continued*

| Reagent type (species) or resource | Designation | Source or reference | Identifiers | Additional information |
|---|---|---|---|---|
| Software, algorithm | cryoSPARC | *Punjani et al., 2017* | RRID:SCR_016501 | https://cryosparc.com/ |
| Software, algorithm | Chimera | *Pettersen et al., 2004* | RRID:SCR_004097 | https://www.cgl.ucsf.edu/chimera |
| Software, algorithm | PyMol | Schrödinger | RRID:SCR_000305 | https://pymol.org/2 |
| Software, algorithm | Coot | *Emsley et al., 2010* | RRID:SCR_014222 | https://www2.mrc-lmb.cam.ac.uk/personal/pemsley/coot/ |
| Software, algorithm | MolProbity | *Chen et al., 2010* | RRID:SCR_014226 | http://molprobity.biochem.duke.edu/ |
| Software, algorithm | PHENIX | *Adams et al., 2010* | RRID:SCR_014224 | https://www.phenix-online.org |
| Other | Superdex 200 Increase 10/300 GL | Cytiva | Cat# 28990944 | |
| Other | Strep-Tactin resin | IBA | Cat# 2-5010 | |
| Other | Amicon Ultra-15 Centrifugal Filter Units | MilliporeSigma | Cat# UFC9100 | |
| Other | Quantifoil R 1.2/1.3 grid Au300 | Quantifoil | Cat# Q37572 | |
| Other | Cellfectin | Thermo Fisher Scientific | Cat# 10362100 | |
| Other | Sf-900 II SFM medium | Thermo Fisher Scientific | Cat# 10902088 | |
| Other | FreeStyle 293 Expression Medium | Thermo Fisher Scientific | Cat# 12338018 | |
| Other | Antibiotic Antimycotic Solution | Sigma-Aldrich | Cat# A5955 | |

## Protein expression and purification

The expression and purification of the HsNCX1 (cardiac isoform NCX1.1, indicated as HsNCX1 or NCX1 throughout the manuscript) were carried out as described previously (*Xue et al., 2023*). Truncated HsNCX1 (Δ341–365aa) containing a C-terminal Strep-tag was cloned into a pEZT-BM vector, and baculoviruses were produced in *Sf9* cells (*Morales-Perez et al., 2016*). For protein expression, cultured Expi293F GnTI- cells were infected with the baculoviruses at a ratio of 1:20 (virus: GnTI-, vol/vol) for 10 hr. 10 mM sodium butyrate was then introduced to boost protein expression level, and cells were cultured in suspension at 30°C for another 60 hr and harvested by centrifugation at 4000×*g* for 15 min. All purification procedures were carried out at 4°C. The cell pellet was resuspended in lysis buffer (25 mM HEPES pH 7.4, 300 mM NaCl, 2 µg/ml DNase I, 0.5 µg/ml pepstatin, 2 µg/ml leupeptin, 1 µg/ml aprotinin, and 0.1 mM PMSF) and homogenized by sonication. NCX1 was extracted with 2% (wt/vol) *N*-dodecyl-β-D-maltopyranoside (DDM, Anatrace) supplemented with 0.2% (wt/vol) cholesteryl hemisuccinate (CHS, Sigma-Aldrich) by gentle agitation for 2 hr, and supernatant collected by centrifugation at 40,000×*g* for 30 min was incubated with Strep-Tactin affinity resin (IBA) for 1 hr. The resin was then collected on a disposable gravity column (Bio-Rad) and washed with 30 column volumes of buffer A (25 mM HEPES pH 7.4, 200 mM NaCl, 0.06% digitonin). NCX1 was eluted in buffer A supplemented with 50 mM biotin and further purified by size-exclusion chromatography on a Superdex 200 10/300 GL column (GE Healthcare). For the generation of NCX1-Fab 2E4 complex, NCX1 was incubated with purified Fab in a molar ratio of 1:1.2 (NCX1: Fab 2E4) for 2 hr and further purified by size-exclusion chromatography in buffer A. The peak fractions were collected and concentrated to ~5–6 mg/ml for cryo-EM analysis. To prepare the protein samples in complex

with various compounds, 0.9 mM SEA0400, 0.47 mM PI(4,5)P$_2$ diC8, or 0.42 mM brain PI(4,5)P$_2$ were added to the protein samples 2 hr before grid preparation.

Expi293F GnTI- cells were purchased from and authenticated by Thermo Fisher Scientific. The cell lines were tested negative for mycoplasma contamination.

## Cryo-EM sample preparation and data acquisition

HsNCX1-Fab 2E4 samples (~5–6 mg/ml) in various conditions were applied to a glow-discharged Quantifoil R1.2/1.3 300-mesh gold holey carbon grid (Quantifoil, Micro Tools GmbH, Germany), blotted for 4.0 s under 100% humidity at 4°C and plunged into liquid ethane using a Mark IV Vitrobot (FEI). For the SEA0400-bound NCX1-Fab 2E4, raw movies were acquired on a Titan Krios microscope (FEI) operated at 300 kV with a K3 camera (Gatan) at 0.83 Å per pixel and a nominal defocus range of –0.9 to –2.2 μm. Each movie was recorded for about 5 s in 60 subframes with a total dose of 60 e$^-$/Å$^2$. For other samples, raw movies were acquired on a Titan Krios microscope operated at 300 kV with a Falcon 4i (Thermo Fisher Scientific) at 0.738 Å per pixel and a nominal defocus range of –0.8 to –1.8 μm. Each movie was recorded for 4 s with a total dose of 60 e$^-$/Å$^2$.

## Image processing

Cryo-EM data were processed following the general scheme described below with some modifications to different datasets (*Figure 2—figure supplements 1 and 2*, *Figure 4—figure supplement 1*). First, movie frames were motion-corrected and dose-weighted using MotionCor2 (*Zheng et al., 2017*). The CTF parameters of the micrographs were estimated using the GCTF program (*Zhang, 2016*). After CTF estimation, micrographs were manually inspected to remove images with bad defocus values and ice contamination. Particles were picked using program Gautomatch (Kai Zhang, https://sbgrid.org/software/titles/gctf) or crYOLO (*Wagner et al., 2019*) and extracted with a binning factor of 3 in RELION (*Zivanov et al., 2018*; *Scheres, 2012*). Extracted particles were subjected to 2D classification, ab initio modeling, and 3D classification. The particles from the best-resolving 3D class were then re-extracted with the original pixel size and subjected to heterogeneous 3D refinement, nonuniform refinement, CTF refinement, and local refinement in cryoSPARC (*Punjani et al., 2017*). The quality of the EM density maps for the TM and cytosolic domains was further improved through focused refinement, allowing for accurate model building for a major part of the protein. For the dataset of NCX1 in complex with brain PI(4,5)P$_2$, the maps of apo inactive NCX1 (PDB 8SGJ) and Ca$^{2+}$-bound active NCX1 (PDB 8SGT) are used as references for heterogeneous refinement. Due to the highly dynamic nature of the protein samples, the particles sorted into the active state produce a map with very low resolution. All resolution was reported according to the gold-standard Fourier shell correlation (FSC) using the 0.143 criterion (*Henderson et al., 2012*). Local resolution was estimated using cryoSPARC.

## Model building, refinement, and validation

The EM maps of HsNCX1 in the SEA0400-bound and PI(4,5)P$_2$ diC8-bound states show high-quality density, and model building is facilitated by previous apo NCX1 structure (PDB 8SGJ) (*Xue et al., 2023*). Models were manually adjusted in Coot (*Emsley et al., 2010*) and refined against maps using the phenix.real_space_refine with secondary structure restraints applied (*Adams et al., 2010*). The final NCX1 structural model contains residues 17–248, 370–467, 482–644, 652–698, 707–718, and 738–935. The EM map of HsNCX1 in complex with brain PI(4,5)P$_2$ is relatively poor. The Ca$^{2+}$-bound activated NCX1 structure (PDB 8SGT) is directly docked into the EM map without adjustment.

The statistics of the geometries of the models were generated using MolProbity (*Chen et al., 2010*). All the figures were prepared in PyMol (Schrödinger, LLC), Chimera (*Pettersen et al., 2004*), and ChimeraX (*Goddard et al., 2018*).

## Electrophysiological experiments

The wild-type HsNCX1 and its mutants were cloned into a pGEMHE vector and expressed in oocytes for electrophysiological recordings. RNA was synthesized using mMessage mMachine (Ambion) and injected into *X. laevis* oocytes as described in *John et al., 2018*. Oocytes were isolated from at least three different frogs and kept at 18°C for 4–7 days. Outward HsNCX1 currents were recorded using the giant patch technique in the inside-out configuration. Each data point shown in this study represents a recording obtained from a single oocyte. The external solution (pipette

solution) contained the following (mM): 100 CsOH (cesium hydroxide), 10 HEPES (4-(2-hydroxyethyl)-1-piperazineethanesulfonic acid), 20 TEAOH (tetraethyl-ammonium hydroxide), 0.2 niflumic acid, 0.2 ouabain, 8 Ca(OH)$_2$ (calcium hydroxide), pH = 7 (using MES, (2-($N$-morpholino) ethanesulfonic acid)); bath solution (mM): 100 CsOH or 100 NaOH (sodium hydroxide), 20 TEAOH, 10 HEPES, 10 EGTA (ethylene glycol-bis(β-aminoethyl ether)-$N$,$N$,$N'$,$N'$-tetra acetic acid) or HEDTA ($N$-(2-Hydroxyethyl) ethylenediamine-$N$,$N'$,$N'$-triacetic acid) and different Ca(OH)$_2$ concentrations to obtain the desired final free Ca$^{2+}$ concentrations, pH = 7 (using MES). Free Ca$^{2+}$ concentrations were calculated according to the WEBMAXc program and confirmed with a Ca$^{2+}$ electrode.

HsNCX1 currents were evoked by the rapid replacement of 100 mM Cs$^+$ with 100 mM Na$^+$, using a computer-controlled 20-channel solution switcher. As HsNCX1 does not transport Cs$^+$, there is no current, and only upon application of Na$^+$ does the exchange cycle initiate. Data were acquired at 4 ms/point and filtered at 50 Hz using an 8-pole Bessel filter. Experiments were performed at 35°C and at a holding potential of 0 mV. The effects of the Na$^+$-dependent inactivation were analyzed by measuring fractional currents calculated as the ratio of the steady-state current to the peak current (fractional activity). All p-values were calculated using an unpaired, two-sided Welch's t-test.

PI(4,5)P$_2$ diC8 (phosphatidylinositol 4,5-bisphosphate diC8, Echelon Bioscience) and brain PI(4,5)P$_2$ (L-α-phosphatidylinositol-4,5-bisphosphate, Brain, Porcine, Avanti Polar Lipids) were dissolved in water and kept as stock at –20°C. Immediately prior to recordings, PI(4,5)P$_2$ was diluted in the bath solution to 10 μM final concentration and perfused cytosolically for the indicated time.

## Acknowledgements

Single particle cryo-EM data were collected at the University of Texas Southwestern Medical Center Cryo-EM Facility, which is funded by the CPRIT Core Facility Support Award RP170644. Cryo-EM sample grids were prepared at the Structural Biology Laboratory at UT Southwestern Medical Center, which is partially supported by grant RP170644 from CPRIT. This work was supported in part by the Howard Hughes Medical Institute (to YJ) and by grants from the National Institute of Health (R35GM140892 to YJ and R01HL152296 to MO) and the Welch Foundation (Grant I-1578 to YJ).

## Additional information

### Funding

| Funder | Grant reference number | Author |
| --- | --- | --- |
| Howard Hughes Medical Institute | | Youxing Jiang |
| National Institute of General Medical Sciences | R35GM140892 | Youxing Jiang |
| National Institutes of Health | R01HL152296 | Michela Ottolia |
| Welch Foundation | I-1578 | Youxing Jiang |

The funders had no role in study design, data collection and interpretation, or the decision to submit the work for publication.

### Author contributions

Jing Xue, Conceptualization, Data curation, Formal analysis, Validation, Investigation, Visualization, Methodology, Writing – original draft, Project administration, Writing – review and editing; Weizhong Zeng, Scott John, Data curation, Formal analysis, Visualization; Nicole Attiq, Data curation, Formal analysis; Michela Ottolia, Conceptualization, Data curation, Formal analysis, Funding acquisition, Validation, Investigation, Writing – original draft, Writing – review and editing; Youxing Jiang, Conceptualization, Resources, Data curation, Formal analysis, Supervision, Funding acquisition, Writing – original draft, Project administration, Writing – review and editing

### Author ORCIDs

Jing Xue (ID) https://orcid.org/0000-0002-7331-1382

Scott John ⓘD https://orcid.org/0000-0002-1232-9140
Michela Ottolia ⓘD https://orcid.org/0000-0002-5114-8887
Youxing Jiang ⓘD https://orcid.org/0000-0002-1874-0504

Reviewer #1 (Public review): https://doi.org/10.7554/eLife.105396.3.sa1
Reviewer #3 (Public review): https://doi.org/10.7554/eLife.105396.3.sa2
Author response https://doi.org/10.7554/eLife.105396.3.sa3

## Additional files

### Supplementary files
MDAR checklist

### Data availability
The cryo-EM density maps of the human NCX1 have been deposited in the Electron Microscopy Data Bank under accession numbers EMD-40456 for the SEA0400-bound state and EMD-60921 for the PI(4,5)P2 diC8-bound state, respectively. Atomic coordinates have been deposited in the Protein Data Bank under accession numbers 8SGI for SEA0400-bound structure and 9IV8 for the PI(4,5)P2 diC8-bound structure. All materials, including plasmids generated in this study, are available from the authors upon request.

The following datasets were generated:

| Author(s) | Year | Dataset title | Dataset URL | Database and Identifier |
|---|---|---|---|---|
| Xue J, Jiang Y | 2024 | Cryo-EM structure of human NCX1 in complex with SEA0400 | https://www.rcsb.org/structure/8SGI | RCSB Protein Data Bank, 8SGI |
| Xue J, Jiang Y | 2025 | Cryo-EM structure of human NCX1 in PIP2 diC8 bound state | https://www.rcsb.org/structure/9IV8 | RCSB Protein Data Bank, 9IV8 |
| Xue J, Jiang Y | 2024 | Cryo-EM structure of human NCX1 in complex with SEA0400 | https://www.ebi.ac.uk/emdb/EMD-40456 | Electron Microscopy Data Bank, EMD-40456 |
| Xue J, Jiang Y | 2025 | Cryo-EM structure of human NCX1 in PIP2 diC8 bound state | https://www.ebi.ac.uk/emdb/EMD-60921 | Electron Microscopy Data Bank, EMD-60921 |

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
