## [Editor Report · eLife Assessment]

Cardiac Ca2+/Na+ exchange is mediated by the NCX1 antiporter, whose activity is tightly regulated. This **important** manuscript describes the structural basis of activation by the lipid DiC8-PIP2 and inhibition by binding of a small molecule to NCX1. These results provide **convincing** insights into NCX1 regulation and the structural basis of cellular Ca2+ signaling.

---

## [Referee Report · Reviewer #1 (Public review)]

This study uses structural and functional approaches to investigate regulation of the Na/Ca exchanger NCX1 by an activator, PIP2 and an inhibitor, SEA0400. Previous functional studies suggest both of these compounds interact with the Na-dependent inactivation process to mediate their effects.

State of the art methods are employed here, and the data are of high quality and presented very clearly. While there is merit in combining structural studies on both compounds as they relate to Na-dependent activation, in the end it is somewhat disappointing that neither is explored in further depth.

The novel aspect of this work is the study on PIP2. Unfortunately, technical limitations precluded structural data on binding of the native PIP2, and so an unnatural short-chained analog, di-C8 PIP2, was used instead. This raises the question of whether these two molecules, which have similar but very distinctly different profiles of activation, actually share the same binding pocket and mode of action. The authors conduct a "competition" experiment, arguing the effect of di-C8-PIP2 addition subsequent to PIP2 suggests competition for a single binding site. In this scenario, PIP2 would need to vacate the binding site prior to di-C8-PIP2 occupying it. However, the lack of an effect of washout alone, suggests PIP2 does not easily unbind. This raises the possibility (probability?) of a non-competitive effect of di-C8-PIP2 at a different site. An additionally informative experiment would be to determine if a saturating concentration of di-C8-PIP2 could prevent the full activation induced by subsequent PIP2 addition. However, the relative affinities of the two ligands might make such an experiment challenging in practice.

In an effort to address the binding site directly, the authors mutate key residues predicted to be important in liganding the phosphorylated head group of PIP2. However, the only mutations that have a significant effect in PIP2 activation also influence the Na-dependent inactivation process independently of PIP2. While these data are consistent with altering PIP2 binding (which cannot be easily untangled from its functional effect on Na-dependent inactivation), a primary effect on Na-inactivation, rather than PIP2 binding, cannot be fully ruled out. A more extensive mutagenic study, based on other regions of the di-C8 PIP2 binding site, would have given more depth to this work and might have been more revealing mechanistically.

The SEA0400 aspect of the work does not integrate particularly well with the rest of the manuscript. This study confirms the previously reported structure and binding site for SEA0400 but provides little further information. While interesting speculation is presented regarding the connection between SEA0400 inhibition and Na-dependent inactivation, further experiments to test this idea are not included here.

Comments on revisions:

(1) The competition assay data for di-C8-PIP2 and PIP2 is a nice addition, but in its description in the text, the authors should be a bit more circumspect about their conclusions, based on the possibility/probability that the effect observed is actually non-competitive (as detailed above).

(2) The authors should acknowledge the formal possibility that the functional effects of the mutations studies are a consequence of a direct effect on Na-dependent inactivation, independent of PIP2 binding.

(3) The authors might strengthen their arguments for combining studies on PIP2 and SEA0400.

(4) The authors could be clearer where their work on SEA0400 extends beyond the previously published observations.

---

## [Referee Report · Reviewer #3 (Public review)]

NCXs are key Ca2+ transporters located on the plasma membrane, essential for maintaining cellular Ca2+ homeostasis and signaling. The activities of NCX are tightly regulated in response to cellular conditions, ensuring precise control of intracellular Ca2+ levels, with profound physiological implications. Building upon their recent breakthrough in determining the structure of human NCX1, the authors obtained cryo-EM structures of NCX1 in complex with its modulators, including the cellular activator PIP2 and the small molecule inhibitor SEA0400. Structural analyses revealed mechanistically informative conformational changes induced by PIP2 and elucidated the molecular basis of inhibition by SEA0400. These findings underscore the critical role of the interface between the transmembrane and cytosolic domains in NCX regulation and small molecule modulation. Overall, the results provide key insights into NCX regulation, with important implications for cellular Ca2+ homeostasis.

Comments on revisions:

The authors have adequately addressed my previous comments.

---

## [Author Response]

The following is the authors’ response to the original reviews

**Reviewer #1 (Public Review):**
(1) This study uses structural and functional approaches to investigate the regulation of the Na/Ca exchanger NCX1 by an activator, PIP2, and an inhibitor, SEA0400. State-of-the-art methods are employed, and the data are of high quality and presented very clearly. The manuscript combines two rather different studies (one on PIP2; and one on SEA0400) neither of which is explored in the depth one might have hoped to form robust conclusions and significantly extend knowledge in the field.

We combined the study of PIP2 and SEA0400 in this manuscript because both ligands inhibit or activate NCX1 by affecting the Na^+^-dependent inactivation of the exchanger - SEA0400 promotes inactivation by stabilizing the cytosolic inactivation assembly whereas PIP2 mitigates inactivation by destabilizing the assembly. The current study aims to provide structural insights into these ligand binding. We didn’t perform extensive electrophysiological analysis as the functional effects of both ligands have been extensively characterized over the last thirty years.

(2) The novel aspect of this work is the study of PIP2. Unfortunately, technical limitations precluded structural data on binding of the native PIP2, so an unnatural short-chained analog, diC8 PIP2, was used instead. This raises the question of whether these two molecules, which have similar but very distinctly different profiles of activation, actually share the same binding pocket and mode of action. In an effort to address this, the authors mutate key residues predicted to be important in forming the binding site for the phosphorylated head group of PIP2. However, none of these mutations prevent PIP2 activation. The only ones that have a significant effect also influence the Na-dependent inactivation process independently of PIP2, thus casting doubt on their role in PIP2 binding, and thus identification of the PIP2 binding site. A more extensive mutagenic study, based on the diC8 PIP2 binding site, would have given more depth to this work and might have been more revealing mechanistically.

The reviewer raises the important question of whether the short-chain PIP2 diC8 and long-chain native PIP2 share the same binding site. We have performed a pilot experiment to address this question. The data indicate that PIP2 diC8 competes with native brain PIP2 for its binding site (Author response image 1). We believe that the mild effects of diC8 on the biophysical properties of NCX1 are due to its decreased affinity as compared to the long-chain PIP2. We have included this competition assay in the revised manuscript.

The acyl-chain length-dependent PIP2 activation is consistent with some previous studies. Before PIP2 was demonstrated to regulate NCX1, some earlier studies showed that negatively charged long-chain lipids such as phosphatidylserine (PS) or phosphatidic acid (PA) could have the same potentiation effects on NCX1 as PIP2 (PMID: 1474504; PMID: 3276350). A later study showed that long-chain acyl-CoAs could also have the same potentiation effects on NCX1 as PIP2 (PMID: 16977318). All these studies demonstrated that activation of NCX by the anionic lipids depends on their chain length with the short chain being ineffective or less effective. These findings have two implications. First, it is the negative surface charge rather than the specific IP3 head group of the lipid that is important for stimulating NCX1 activity. This would imply non-specific electrostatic interactions between the negatively charged lipids and those positively charged residues at the binding site. Second, a longer acyl chain is required for the high-affinity binding of PIP2 or negatively charged lipids. As further discussed in the revised manuscript (Discussion section), we suspect the tail of the long acyl chain from the native anionic lipids can enter the same binding pocket for SEA0400 thereby rendering higher affinity lipid binding than shorter chain lipids.

As the interactions between PIP2 and NCX1 are both electrostatic involving multiple charged residues as well as hydrophobic involving the long lipid acyl chain, single amino acid substitutions likely only decrease the affinity of PIP2 rather than completely disrupt its binding. Our data demonstrated that mutants R220A, K225A, and R220A/K225A do show a significantly decreased potentiation effect of PIP2 (Figure 3 in the manuscript). We also conducted an experiment with a mutant exchanger in which all four amino were mutated. This K164A/R167A/R220A/K225A mutant is insensitive to PIP2 and shows no Na^+^-dependent inactivation (Figure 3A). The unresponsiveness to PIP2 and lack of Na^+^-dependent inactivation in this mutant is consistent with previous studies demonstrating that PIP2 activates NCX by tuning the amount of Na^+^-dependent inactivation and any mutation that decreases NCX sensitivity to PIP2 will affect the extent of Na^+^-dependent inactivation (PMID: 10751315). Such studies show that the two processes cannot be dissected from each other, making more extensive mutagenesis investigation unlikely to provide new mechanistic insights. A brief discussion related to this quadruple mutant has been added in the revised manuscript.

**Author response image 1. sa3fig1:** Giant patch recording of the human WT exchanger. Currents were first activated by intracellular application of 10 µM brain PIP2. Afterwards, a solution containing 100 mM Na^+^ and 12 µM Ca^2+^ was perfused for about 5 min (washout). The PIP2 effects was not reversible during this time. The same patch was then perfused internally with the same solution in presence of 10 µM di-C8. Application of the shorted-chained di-C8, partially decreased the current suggesting that that PIP2 and diC8 compete for the binding site.

(3) The SEA0400 aspect of the work does not integrate particularly well with the rest of the manuscript. This study confirms the previously reported structure and binding site for SEA0400 but provides no further information. While interesting speculation is presented regarding the connection between SEA0400 inhibition and Na-dependent inactivation, further experiments to test this idea are not included here.

Our SEA0400-bound NCX structure was determined and deposited in 2023, along with our previous study on the apo NCX published in 2023 (PMID: 37794011). We decided to combine the SEA0400-bound structure with the later study of PIP2 binding because both represent ligand modulation of NCX by affecting the Na^+^-dependent inactivation of the exchanger. The SEA0400 inhibition of NCX1 has been extensively investigated previously, which demonstrated a strong connection between SEA0400 and the Na^+^-dependent inactivation. As discussed in the manuscript, SEA0400 is ineffective in an exchanger lacking Na^+^-dependent inactivation. Conversely, enhancing the extent of Na^+^-dependent inactivation increases the affinity for SEA0400. Our structural analysis provides explanations for these pharmacological features of SEA0400 inhibition.

**Reviewer #2 (Public review):**
(1) The study by Xue et al. reports the structural basis for the regulation of the human cardiac sodium-calcium exchanger, NCX1, by the endogenous activator PIP2 and the small molecule inhibitor SEA400. This well-written study contextualizes the new data within the existing literature on NCX1 and the broader NCX family. This work builds upon the authors' previous study (Xue et al., 2023), which presented the cryo-EM structures of human cardiac NCX1 in both inactivated and activated states. The 2023 study highlighted key structural differences between the active and inactive states and proposed a mechanism where the activity of NCX1 is regulated by the interactions between the ion-transporting transmembrane domain and the cytosolic regulatory domain. Specifically, in the inward-facing state and at low cytosolic calcium levels, the transmembrane (TM) and cytosolic domains form a stable interaction that results in the inactivation of the exchanger. In contrast, calcium binding to the cytosolic domain at high cytosolic calcium levels disrupts the interaction with the TM domain, leading to active ion exchange.In the current study, the authors present two mechanisms explaining how both PIP2 stimulates NCX1 activity by destabilizing the protein's inactive state (i.e., by disrupting the interaction between the TM domain and the cytosolic domain) and how SEA400 stabilizes this interaction, thereby acting as a specific inhibitor of the system.The first part of the results section addresses the effect of PIP2 and PIP2 diC8 on NCX1 activity. This is pertinent as the authors use the diC8 version of this lipid (which has a shorter acyl chain) in their subsequent cryo-EM structure due to the instability of native PIP2. I am not an electrophysiology expert; however, my main comment would be to ask whether there is sufficient data here to characterise fully the differences between PIP2 and PIP2 diC8 on NCX1 function. It appears from the text that this study is the first to report these differences, so perhaps this data needs to be more robust. The spread of the data points in Figure 1B is possibly a little unconvincing given that only six measurements were taken. Why is there one outlier in Figure 1A? Were these results taken using the same batch of oocytes? Are these technical or biological replicates? Is the convention to use statistical significance for these types of experiments?

Oocytes were isolated from at least 3 different frogs and each data point shown in Fig. 1 A or 1B of the manuscript represents a recording obtained from a single oocyte. For clarity, we have added this information to the Methods section. We understand that 6 observations (Fig. 1B) are a small sample size but electrophysiological recordings of NCX currents are extremely challenging and technically difficult due to the low transport activity of the exchanger. Because of these circumstances, this type of study relies on a small sample of observations. Nevertheless, our data clearly show that native PIP2 and the short-chain PIP2 diC8 can activate NCX activity although with different affinity. The spread of the steady state current data points is due to the variability in the extent of Na^+^-dependent inactivation within each patch, likely due to slightly different levels of endogenous PIP2 or other regulatory mechanisms that control this allosteric process. As PIP2 acts on the Na^+^-dependent inactivation this will lead to varying levels of potentiation. Because of that, we did occasionally observe some outliers in our recordings. Rather than cherry-picking in data analysis, we presented all the data points from patches with measurable NCX1 currents. Despite this variability, a T-test indicates that the effects of PIP2 are more pronounced on the steady-state current than peak current. The differences between native PIP2 and PIP2 diC8 on NCX1 function are consistent with previous investigations showing that both PIP2 and anionic lipids enhance NCX current by antagonizing the Na^+^-dependent inactivation and long-chain lipids are more effective in potentiating NCX1 activity (PMID: 1474504; PMID: 3276350; PMID: 16977318). A discussion related to the chain length-dependent lipid activation of NCX1 is added in the Discussion of the revised manuscript.

(2) I am also somewhat skeptical about the modelling of the PIP2 diC8 molecule. The authors state, "The density of the IP3 head group from the bound PIP2 diC8 is well-defined in the EM map. The acyl chains, however, are flexible and could not be resolved in the structure (Fig. S2)."However, the density appears rather ambiguous to me, and the ligand does not fit well within the density. Specifically, there is a large extension in the volume near the phosphate at the 5' position, with no corresponding volume near the 4' phosphate. Additionally, there is no bifurcation of the volume near the lipid tails. I attempted to model cholesterol hemisuccinate (PDB: Y01) into this density, and it fits reasonably well - at least as well as PIP2 diC8. I am also concerned that if this site is specific for PIP2, then why are there no specific interactions with the lipid phosphates? How can the authors explain the difference between PIP2 and PIP2 diC8 if the acyl chains don't make any direct interactions with the TM domain? In short, the structures do not explain the functional differences presented in Figure 1.The side chain densities for Arg167 and Arg220 are also quite weak. While there is some density for the side chain of Lys164, it is also very weak. I would expect that if this site were truly specific for PIP2, it should exhibit greater structural rigidity - otherwise, how is this specific?Given this observation, have the authors considered using other PIP2 variants to determine if the specificity lies with PI4,5P_2_ as opposed to PI3,5P_2_ or PI3,4P_2_? A lack of specificity may explain the observed poor density.

The map we provided to the editor in the initial submission is the overall map for PIP2-bound NCX1. Due to the relative flexibility between the cytosolic CBD and TM regions, we also performed local refinement on each region in data processing to improve the map quality as illustrated in Fig. S2. The local-refined map focused on the TM domain provides a much better density for PIP2 diC8 and its surrounding residues than the overall map. The map quality allowed us to unambiguously identify the lipid as PIP2 with the IP3 head group having phosphate groups at the 4,5 positions. Furthermore, no lipid density is observed at the equivalent location in the local-refined map from the apo NCX1 TM region as shown in Fig. S3 in the revision. In the revised manuscript, the density for the bound PIP2 is shown in Fig. 2A. Those local-refined maps for PIP2-bound NCX1 were also deposited as additional maps along with the overall map in the Electron Microscopy Data Bank under accession numbers EMD-60921. The local-refined maps for the apo-NCX1 were deposited in the Electron Microscopy Data Bank under accession numbers EMD-40457 in our previous study (https://www.ebi.ac.uk/emdb/EMD-40457?tab=interpretation).

As discussed in our response to reviewer #1, the acyl-chain length-dependent PIP2 activation is consistent with some previous studies. Before PIP2 was identified as a physiological regulator of NCX1, some earlier studies showed that negatively charged long-chain lipids such as phosphatidylserine (PS) or phosphatidic acid (PA) could have the same potentiation effects on NCX as PIP2 (PMID: 1474504; PMID: 3276350). A later study also showed that acyl-CoA could also have the same potentiation effects on NCX as PIP2 (PMID: 16977318). All these studies demonstrated that activation of NCX1 by the anionic lipids depends on their chain length with the short chain being ineffective. These findings have two implications. First, it is the negative surface charge rather than the specific IP3 head group of the lipid that is important for stimulating NCX activity. This would imply non-specific electrostatic interactions between the negatively charged lipids and those positively charged residues at the binding site. Second, a longer acyl chain is required for the high-affinity binding of PIP2 or negatively charged lipids. As further discussed in the revised manuscript (Discussion section), we suspect the tail of the long acyl chain can enter the same binding pocket for SEA0400 thereby rendering higher affinity lipid binding than shorter chain lipids. In light of the equivalent potentiating effect of various anionic lipids on NCX1, PI(4,5)P2 activation of NCX1 is likely non-specific and PI(3,5)P2 or PI(3,4)P2 may also activate the exchanger. However, as a key player in membrane signaling, PI(4,5)P2 has been demonstrated to be a physiological regulator of NCX1 in many studies.

(3) I also noticed many lipid-like densities in the maps for this complex. Is it possible that the authors overlooked something? For instance, there is a cholesterol-like density near Val51, as well as something intriguing near Trp763, where I could model PIP2 diC8 (though this leads to a clash with Trp763). I wonder if the authors are working with mixed populations in their dataset. The accompanying description of the structural changes is well-written (assuming it is accurate).

Densities from endogenous lipids and cholesterols are commonly observed in membrane protein structures. Other than the bound PIP2, those lipid and cholesterol densities are present in both the apo and PIP2-bound structures, including the density around Trp763 and Val53. Whether those bound lipids/cholesterols play any functional roles or just stabilize the protein is beyond the scope of this study. We have added a supporting figure (Fig. S3) showing a side-by-side comparison of the density at the PIP2 binding site between the PIP2-bound and apo structures.

I would recommend that the authors update the figures associated with this section, as they are currently somewhat difficult to interpret without prior knowledge of NCX architecture. My suggestions include:- Including the density for the PIP2 diC8 in Figure 2A.

As suggested, we have included the density of PIP2 diC8 in Figure 2A.

- Adding membrane boundaries (cytosolic vs. extracellular) in Figure 2B.- Labeling the cytosolic domains in Figure 2B.- Adding hydrogen bond distances in Figure 2A.

We have added and labeled the boundaries for the TM and cytosolic domains in Figure 2B as suggested. Although we can identify those positively charged residues in the vicinity of the PIP2 head group and observe local structural changes, the poorly defined side-chain densities of these residues won’t allow us to properly determine the hydrogen bond distances.

- Detailing the domain movements in Figure 2B (what is the significance of the grey vs. blue structures?).

There is a rigid-body downward swing movement at CBDs between the apo (grey) and PIP2-bound (cyan) structures. The movement at the TM region is subtle. We have added the description in the legend for Figure 2B and also marked the movement at the tip of CBD1 in the figure.

The section on the mechanism of SEA400-induced inactivation is strong. The maps are of better quality than those for the PIP2 diC8 complex, and the ligand fits well. However, I noticed a density peak below F02 on SEA400 that lies within the hydrogen bonding distance of Asp825. Is this a water molecule? If so, is this significant?

The structure of SEA0400-bound NCX1 was determined at a higher resolution likely because the drug stabilize the exchanger in the inactivated state. The mentioned density could be an ordered water molecule. We don’t know if it is functionally significant.

Furthermore, there are many unmodeled regions that are likely cholesterol hemisuccinate or detergent molecules, which may warrant further investigation.

We constantly observed partial densities from bound lipids, cholesterols, or detergents in our structures. Most of them are difficult to be unambiguously identified and modeled. Whether they play any functional roles is beyond the scope of this study.

The authors introduce SEA400 as a selective inhibitor of NCX1; however, there is little to no comparison between the binding sites of the different NCX proteins. This section could be expanded. Perhaps Fig. 4C could include sequence conservation data.

SEA0400 is more specific for NCX1 than NCX2 and NCX3 as demonstrated in an early study (PMID: 14660663). The lack of structure information for NCX2 or NCX3 makes it difficult to make a direct comparison to reveal the structural basis of SEA0400 specificity.

Additionally, is the fenestration in the membrane physiological, or is it merely a hole forced open by the binding of SEA400? I was unclear as to whether the authors were suggesting a physiological role for this feature, similar to those observed in sodium channels.

The fenestration likely serves as the portal for SEA0400 binding as discussed in the manuscript. As further discussed in the revised manuscript, we suspect this fenestration also allows the tail of a long-chain lipid to enter the same binding pocket for SEA0400 and results in higher affinity binding of a long-chain lipid than a short-chain lipid.

**Reviewer #3 (Public review):**
NCXs are key Ca^2+^ transporters located on the plasma membrane, essential for maintaining cellular Ca^2+^ homeostasis and signaling. The activities of NCX are tightly regulated in response to cellular conditions, ensuring precise control of intracellular Ca^2+^ levels, with profound physiological implications. Building upon their recent breakthrough in determining the structure of human NCX1, the authors obtained cryo-EM structures of NCX1 in complex with its modulators, including the cellular activator PIP2 and the small molecule inhibitor SEA0400. Structural analyses revealed mechanistically informative conformational changes induced by PIP2 and elucidated the molecular basis of inhibition by SEA0400. These findings underscore the critical role of the interface between the transmembrane and cytosolic domains in NCX regulation and small molecule modulation. Overall, the results provide key insights into NCX regulation, with important implications for cellular Ca^2+^ homeostasis.

We appreciate this reviewer’s positive comments.

**Recommendations for the authors:**

**Reviewer #1 (Recommendations for the authors):**
The manuscript would be strengthened enormously by a much deeper focus on the novel and very interesting PIP2 work, as noted above, and perhaps the removal of the SEA0400 data.If that is beyond the scope of the authors' options, then a more robust discussion of limitations of the current work, perhaps speculation regarding other future experiments, a clearer presentation of how these data on SEA0400 are different from/extend from the previously published work, and a better effort to link the two disparate aspects of the work into a more cohesive manuscript should be attempted.

As discussed in our response to this reviewer’s public review, we combined the study of PIP2 and SEA0400 in this manuscript because both ligands activate or inhibit NCX1 by affecting the Na^+^-dependent inactivation of the exchanger. The functional effects of both ligands on NCX1 have been extensively characterized over the last thirty years. Thus the current study is focused on providing structural explanations for some unique pharmacological features of these ligands. In the revised manuscript, we have added an extra paragraph of discussion that provides a plausible explanation for chain length-dependent PIP2 activation.

**Reviewer #3 (Recommendations for the authors):**
A few comments to consider:(1) The short-chain PIP2 appears to have lower potency, but the mechanism remains unclear. Based on structural analyses, are there potential binding sites for the acyl chains of PIP2 that could contribute to this difference?

As discussed in our response to other reviewers, long-chain anionic lipids can have the same potentiation effect on NCX1 activity as PIP2, but the short-chain ones are ineffective just like short-chain PIP2 diC8. We suspect the tail of a long acyl chain from the native PIP2 can enter the same binding pocket for SEA0400 thereby rendering higher affinity binding for a long-chain lipid than a short-chain lipid. A discussion related to this point has been added to the revised manuscript.

(2) It is unclear why mutating residues that interact with the IP3 head group retain PIP2 activation. Would it be possible to assess PIP2 and C8 PIP2 binding to these NCX1 variants? Identifying a mutant that abolishes C8 PIP2 binding would be valuable in interpreting those results.

As the interactions between PIP2 and NCX1 are both electrostatic involving multiple charged residues and hydrophobic involving the long lipid acyl chain, single amino acid substitutions likely only decrease the affinity of PIP2 rather than completely disrupt its binding. Individual mutants R220A and K225A show a 5-fold decrease in their response to PIP2 application indicating that their replacement alters the affinity of NCX for PIP2. We have added a new experiment showing that an exchanger with all four residues mutated is insensitive to PIP2 in the revision.

(3) What are the functional effects of mutating Y226 and R247, residues that seem to play an important role in PIP2-mediated activation?

In a previous study, mutation at Y226 (Y226T), which is found within the XIP region of NCX, has been shown to have enhanced Na^+^-dependent inactivation (PMID: 9041455). To our knowledge, the R247 mutation has not been investigated. Also positioned in the XIP region, we suspect its mutation could directly affect Na^+^-dependent inactivation. This would make it difficult to determine if the function effect of the mutation is caused by changing the stability of the XIP region or by changing the binding of PIP2.

(4) Is there any overlap between the PIP2 and SEA0400 binding regions? Both appear to involve TM4, TM5, and TMD-beta hub interfaces. It might be interesting to discuss any shared mechanisms and why this region might serve as a hotspot for modulation.

As mentioned in our previous response, we suspect the tail of a long acyl chain from the native PIP2 can enter the same binding pocket for SEA0400 thereby rendering higher affinity binding for a long-chain lipid than a short-chain lipid. A more detailed discussion related to this point has been included in the revision.

(5) It would be helpful to show the density at the PIP2-binding site in the apo and PIP2-bound structures side by side

This figure has been added in the revision as Fig. S3.